# The ITransverse.jl library for transverse tensor network contractions

**Stefano Carignano**[1⋆],

**1** Barcelona Supercomputing Center, 08034 Barcelona, Spain

⋆ stefano.carignano@bsc.es ,

## Abstract

Transverse contraction methods are extremely promising tools for the efficient contraction of tensor networks associated with the time evolution of quantum many-body systems, allowing in some cases to circumvent the entanglement barrier that would normally prevent the study of quantum dynamics with classical resources. We present here the ITransverse.jl package, written in Julia and based on ITensors.jl, containing several of these high-level algorithms, including novel prescriptions for efficient truncations of temporal matrix product states.

# 1   Introduction

The study of the dynamics of quantum many-body systems is a fascinating yet extremely challenging subject in contemporary physics. Several open questions on the topic, such as the nature of thermalization or the possible breakdown of ergodicity and the formation of many-body localized phases [1, 2] that retain memory of their initial conditions, require a detailed characterization of the time evolution of quantum systems, a task which -from the theoretical side- becomes unfeasible using traditional tools due to the exponential cost in representing wave-functions for many-particle systems.

Over the past few years, theoretical condensed matter physics has been revolutionized by the development of tensor network (TN) algorithms, which have allowed to characterize the equilibrium properties of quantum many-body systems with unprecedented accuracy. Nowadays, methods based on the Density Matrix Renormalization Group (DMRG) [3] formulated in the language of Matrix product states (MPS) are the gold standard for the study of lower-dimensional quantum many-body systems, and a very active research field has developed around these techniques (for reviews see eg. [4–6]). In this work, we will mostly focus on such lower-dimensional systems, particularly 1D spin chains, though many of the techniques discussed in the following can in principle be applied to higher-dimensional systems as well.

Traditionally, the expressivity of MPS as a wave-function ansatz for a quantum system is related to entanglement [7, 8], which provides a basis on which an efficient compression of quantum states can be performed [9]. Since ground states of one-dimensional gapped local Hamiltonians are known to satisfy an area law for their entanglement, this implies that they can be efficiently represented using MPS, and optimized by performing local operations with a computational cost that scales polynomially (instead of exponentially) with the number of constituents of the system [10]. Since "area law" implies that the entanglement of a bipartition of the system in two parts is proportional to the area of the boundary between them, this means that in a one-dimensional system (where the area between any two segments is just two points) the entanglement is bounded and does not grow arbitrarily with the size of the system. A very different scenario would be given in the case of a "volume law", for which the entanglement of a 1D chain would grow arbitrarily large as the number of its elements increases.

From a technical point of view, having a limited amount of entanglement results in a small virtual (often called "bond") dimension of the tensors forming the MPS, allowing for an efficient compression of the data required to faithfully represent these states. Other well-known examples of efficient representations using tensor networks are thermal states, which can be described by density operators, as well as local Hamiltonians (and their exponentials), which allow for a compressed matrix product operator (MPO) form [11–15].

Given the tremendous success of these methods in describing ground state and equilibrium properties of quantum many-body systems, it is then logical to ask whether they can provide an efficient representation for studying their time evolution as well. After all, if a system thermalizes, a typical scenario such as a quench from a ground state to another thermal state would interpolate between two limits characterized both by low entanglement.

The answer in this case is less clear: while tremendous progress has been made in developing highly efficient algorithms based on a Trotterized time evolution such as TEBD [16], tDMRG [17] and the like, as well as variational methods which directly solve the Schrödinger equation on the tangent space of these MPS [18], ultimately they all rely on a final re-compression of the wave-function which, as before, will depend on the amount of entanglement between the constituents of the system (see [19] for a comprehensive review). Unfortunately, it is nowadays well understood that the entanglement entropy of a wave function in a standard time evolution scenario such as a global quench grows linearly with time [20, 21], requiring in principle an exponential amount of resources to faithfully describe the state in time. This so-called "entanglement barrier" constitutes a formidable challenge for any current TN method based on constructing the time-evolved wave function for a many-body system.

In recent years, new tensor network algorithms for studying time evolution have nevertheless been proposed. One extremely promising example is given by *transverse contraction* methods [22]: the idea is to encode the dynamical evolution of a $D$-dimensional system in a $D + 1$-dimensional tensor network, where the additional dimension is given by time, and contract it along the spatial direction using boundary MPS methods (see the following Section). Improving over the first proposals in this direction, we proposed a novel truncation method based on reduced *transition* matrices, which allowed to develop further intuition on the computational complexity of computing the dynamics of quantum many-body systems using tensor networks [23]. This prescription also provides a natural connection with concepts introduced in field theory such as generalized temporal entropies [23–25], the physics of open quantum systems [26] and, importantly, allows to compute expectation values of local operators efficiently [27].

During the course of these investigations held by our joint collaboration at BSC, U. Barcelona and CSIC[1] we began developing a toolkit implementing transverse contraction algorithms, most of which were lacking a public code implementation. The process culminated with the creation of the `ITransverse.jl` package, written in the modern Julia programming language [28] and built on top of the `ITensors.jl` and `ITensorMPS.jl` libraries [29], which provide an easy way to write performant code that can be straightforwardly extended to use GPU acceleration. The idea is to provide a self-contained implementation of most state of the art algorithms related to transverse contraction, which allows the user to implement their favorite model and compute in the most efficient way possible time-dependent amplitudes, expectation values and (generalized) entropies.

Installation is simply done via Julia's package manager:

```julia
julia> using Pkg
julia> Pkg.add(url="https://github.com/starsfordummies/ITransverse.jl.git")
```

Before moving on to the description of the algorithms included in the library, let us briefly review the ideas behind these transverse contraction algorithms for studying the time evolution of quantum many-body systems.

---

[1]The people who actively contributed to this line of research are Luca Tagliacozzo, Carlos Ramos Marimón, Aleix Bou-Comas, Jan T. Schneider, Esperanza López and Sergio Cerezo.

## 2   Time evolution and transverse contraction methods

The most commonly employed tensor network methods for studying the time evolution of a quantum many-body system, such as TEBD and TDVP, focus on building the time-evolved wave-function of the system in a compressed form, corresponding to the familiar Schrödinger picture in quantum mechanics. As already anticipated in the previous section, this approach often turns out to be computationally inefficient, as during time evolution entanglement tends to grow with a volume law, rendering an efficient re-compression using MPS impossible.

As is well known, focusing on the time evolution the wave function however is not the only way to perform time evolution in quantum mechanics. Under many circumstances, we are mainly interested in computing expectation values of a (typically local) operator $O$, say for example the magnetization for a spin:

$$\langle O(t) \rangle = \langle \psi(0)|U(t)^{\dagger}OU(t)|\psi(0) \rangle \tag{1}$$

where $|\psi(0)\rangle$ denotes the initial state and $U(t) = \exp(-iHt)$ is the time evolution operator.

Since both the operator and the time-evolution operator for a local Hamiltonian $H$ can usually be efficiently expressed in form of matrix product operators (MPOs) [12], another natural option would be to try and build a time-evolved $O(t) = U(t)^{\dagger}OU(t)$, which is nothing but the usual Heisenberg picture for time evolution in quantum mechanics. While this approach turns out to be computationally advantageous in a few cases, in a generic scenario the "operator entanglement" which dictates the bond dimension required for an efficient representation of $O(t)$ also turns out to grow linearly with time, so that once again the entanglement barrier prevents from accessing long-time evolution due to an exponential growth in the computational resources required [30, 31].

It may appear then that performing long-time evolution with tensor networks is a hopeless task due to the growth of entanglement between the sites of the system, either on the operator or the state side, as time increases.

As we will see in the following, tensor networks can however hint towards an additional way to perform time evolution beyond the traditional Schrödinger and Heisenberg pictures. The key idea for this is to visualize the dynamical evolution of a $D$-dimensional system as a $D + 1$-dimensional tensor network, where the additional dimension is given by time. Up to Trotter errors, this higher-dimensional PEPS of finite bond dimension then encodes the full dynamics we are interested in; the challenge is now to contract it efficiently.

We can start for example by inspecting the two-dimensional TN associated with the calculation of a return amplitude of a product state $|\psi_0\rangle$ to itself after some time evolution with a given Hamiltonian, which is often referred to as a Loschmidt echo (see Fig. 1),

$$\mathcal{A}_{\psi_0\psi_0}(t) = \langle \psi(0)|U(t)|\psi(0) \rangle \,. \tag{2}$$

The basic ingredients of the network here are the tensors of the initial and final states, represented as an MPS, and the MPO tensors that make the time evolution operator $U(t)$. For simplicity, in the following we shall mostly work with translation invariant systems and time-independent Hamiltonians, so that all these MPO tensors $W$ in the bulk are identical.

Now, instead of evolving the initial product state by contracting rows of the $U(t)$ MPO onto it and building up entanglement among its sites, we can think of contracting this network in a *transverse* direction, namely by identifying columns of the TN with states and operators associated with one spatial site at different time steps [22,32,33]. The columns at the left and right edges of the network will thus now represent a *temporal* matrix product state (tMPS), whereas the columns in the middle, which act as spatial transfer matrices (we will usually refer to them as $E$, as in Figure 1) can be seen as temporal MPOs (tMPO). We will go over the details on the construction of these objects in Section 3.

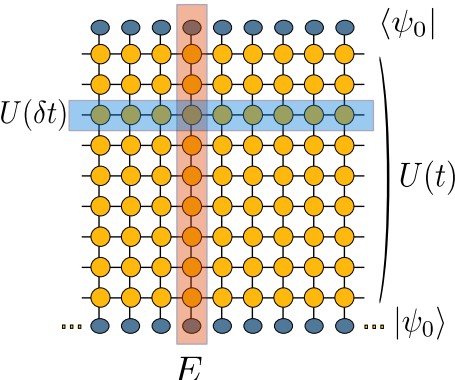

Figure 1: Two-dimensional (space-time) tensor network representing a time-dependent return amplitude $\langle\psi_0|U(t)|\psi_0\rangle$: starting from $|\psi_0\rangle$ product state, which we draw at the bottom of the network (ie. time runs upwards), we write the time-evolution operator $U(t)$ as the product of several rows of MPOs, each one associated with the trotterized $U(\delta t)$ (shaded in blue). We then close again with the conjugate of the initial state, represented at the top. For a translation-invariant system, the whole network is given by the repeated product of a single column $E$, which can be seen as a transfer matrix that moves from one spatial site to the next.

Starting from the left and right edges, we can apply the standard TN machinery to perform the contraction of the network by applying the tMPOs to the tMPS at the edges. Naturally, the bond dimension of the resulting tMPS can in principle grow exponentially with the number of tMPO layers applied to them, so that, as usual, some intermediate compression is required (we will discuss this issue in detail in Section 4).

We are then back to the question of whether these temporal states can be efficiently re-compressed in MPS form at each step. If we turn to the standard DMRG-type algorithms, the determining factor here will be a "temporal" entanglement, associated with a given element of the system at different times [22, 32–34]. While the physical interpretation of this quantity is still object of an active investigation, from a computational point of view this is a well-defined question: one just computes the standard entanglement entropy along bipartitions of the tMPS via its reduced density matrices and checks its scaling with the number of sites, which now coincides with the number of steps $N_t$ of time evolution performed: that is, longer chains correspond to longer time evolutions.

This prescription was first applied in [22] to compute the expectation value of a local operator, cf. Equation (1). The tensor network associated with this quantity is shown in Figure 2 (a): half of the rows are given by the time evolution $U(t)$ MPOs, whereas the other half by its conjugate $U^\dagger(t)$.

Following intuition from simpler toy models, in order to reduce temporal entanglement the authors of [22] suggested then to *fold* the tensor network in half along the temporal direction, building composite tensors made of an element of the forward and one of the backwards evolution MPO (Figure 2 (b)). This construction is reminiscent of the forwards-backwards Schwinger-Keldysh contour, which is used in real-time field theory calculations. The folding prescription also allows to build a connection with familiar concepts such as the Feynman-Vernon influence functional, as pointed out in [34, 35]: focusing on one constituent, we can interpret the left and right vectors built from the contraction to the rest of the network to its left (or right, respectively) as the description of the "bath", represented by the rest of the system, at different times. The ability to consider the full spatio-temporal network here can allow us to go even further, identifying temporal MPS as objects which can be used to represent

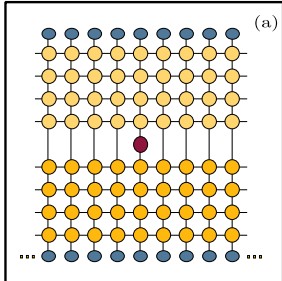 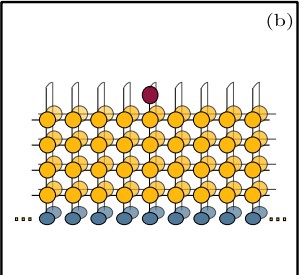 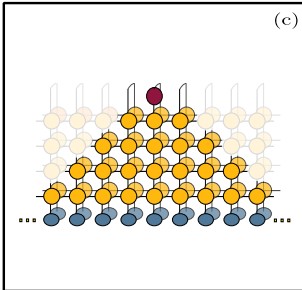

Figure 2: (a) Two-dimensional tensor network associated with the expectation value of a local operator (denoted as a red tensor in the middle of the network. (b) Folded version of the same network: in our convention, the vectorized local operator is at the top of the network, while the initial state (or rather density matrix, in the folded picture) is at the bottom. (c) Light cone structure: via the folding operation, all tensors outside of the causal cone of the local operators reduce to identities, so we can neglect them in the actual computations.

*process* tensors, allowing for a direct description of correlations in space and time in the unified language of tensor networks [26].

The folded structure for an expectation value of a local operator allows also to exploit the finite speed of propagation of information in the system. By building the TN in a form which reproduces the light cone associated with the operator, one can effectively reduce the number of tensors required for evaluating its expectation value: all information outside the cone will not play a role in the dynamics. Specifically, one can see that, upon contraction, all folded tensors outside the cone collapse to identities (see Figure 2 (c)).

Having defined the boundary tMPS and tMPO, we are now ready to build algorithms that contract the TN using them. In the `ITransverse.jl` library we have implemented both a power method, which can be used to build left and right vectors for systems with an infinite spatial extension, as well as a light-cone method tailored for the evaluation of expectation values of local operators, which exploits the causal structure discussed above. We discuss these methods in detail in Section 5.

While all these novel techniques turn out to be advantageous in several situations compared to traditional methods, there are still cases in which temporal entanglement exhibits a volume law, ie. increases linearly with the time of the evolution considered, effectively preventing their use for long time evolution.

A closer inspection of the time-evolution TN with a mindset focused on transverse contractions can nevertheless give us even more information. The key observation here is that, if we consider the contraction of the network starting from the sides and construct a tMPS $\langle L|$ associated with the contraction of the left half of the system, as well as a "right" tMPS $|R\rangle$ for the other half, the final result of our calculation will be given by the overlap between these two, $\langle L|R\rangle$. The relevant object for our calculation then is not given by the left and right vectors separately, but rather by their overlap, upon which one can construct a cost function to optimize when performing a truncated contraction of the network [23]. In turn, this implies that - rather than thinking about the usual (reduced) density matrices (RDMs) $\rho_L = |L\rangle\langle L|$ and $\rho_R = |R\rangle\langle R|$ for left and right vectors individually, one should consider *transition* matrices $\mathcal{T} \propto |R\rangle\langle L|$ and their reduced (RTMs).

Aside from providing a possible computational advantage, the method proposed in [23] allows for a fascinating connection with high energy physics: there the concept of reduced transition matrices has been recently proposed as starting point to defined *generalized* en-

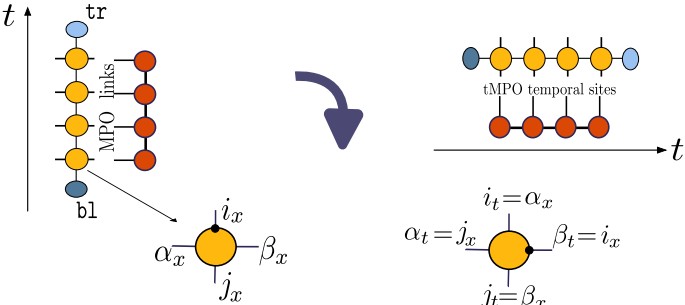

Figure 3: Sketch of the rotation implicitly performed in `ITransverse.jl` for index labelling: the physical indices $(i_x, j_x)$ of the usual tensors of the MPO $U(\delta t)$ become virtual indices $(\alpha_t, \beta_t)$ for the temporal MPO, whereas the virtual indices $(\alpha_x, \beta_x)$ become physical temporal ones $(i_t, j_t)$. The physical Hilbert space of the temporal degrees of freedom is thus dictated by the virtual dimension of $U(\delta t)$. Each tMPO column is made by a stack of rank-4 tensors closed at the top and the bottom (or right and left, after rotation) by the `tr` and `bl` states.

tropies [36–38], which turn out to have a geometrical interpretation in holography [38–42]. For our case, the relevant quantity in the optimization would then be the *generalized temporal entropy* associated with the left-right contraction of our network. Since the RTMs are not hermitian by construction, in principle these entropies are complex quantities, so that their interpretation has to be taken with care. We will discuss the calculation of these entropies, their properties and their relation with the computational complexity of contracting the tensor network in Section 6.

In the remainder of this work, we will review these aspects associated with transverse contraction algorithms, exploring along the way the various functionalities implemented in the `ITransverse.jl` library.

## 3 Temporal MPOs and transverse contraction

Much like for traditional time-evolution methods such as TEBD, the basic building blocks for all the algorithms discussed here are the tensors of a matrix product operator, which can be built in the usual way by factorizing the Trotterized time evolution operator $U(\delta t)$ as a product of local tensors. In order to perform the transverse contraction, we identify *columns* of these tensors, defining temporal MPOs (tMPOs) $E$, as shown in Figure 1. In the same way, at the boundaries of the network we identify left and right temporal MPS, and contract the network using these boundary states. In this transverse picture, the links of the time evolution MPO thus become associated with a Hilbert space (with physical dimension given by the bond dimension of $U(\delta t)$) where temporal degrees of freedom live, while the original physical legs become links in the temporal direction.

In order to build a connection with the usual notation employed by TN libraries (and to most people's intuition), where we think of MPS as chains extending from left to right, in the following we will often work in a "rotated" picture, namely we consider the network as tilted clockwise by 90 degrees. The top side of the tMPS/tMPO then becomes the right edge of the rotated chains. We illustrate this in Figure 3. Of course, this rotation is merely a visualization tool (so that we can, eg., talk about left and right canonical forms), which has no effect on the physics underneath.

We start in Section 3.1 describing the basic building blocks of the networks we consider.

Figure 4: (a): Given a two-body gate such as those that typically appear in the Trotter decomposition of exponentials of local Hamiltonians, we can decompose them using standard tensor factorizations. (b): The typical Trotter structure for $U(\delta t)$ made by the product of even-odd terms can be brought into MPO form after decompositions and contractions of pairs of the resulting tensors (denoted as shaded regions). Assuming translation-invariant Hamiltonians, for a open chain of $N$ sites, we end up with $N-2$ bulk tensors $W_c$ sandwiched between a left $W_l$ and a right $W_c$ boundary tensors.

As described in the previous section, when dealing with dynamics one can typically encounter two types of scenario: in the first one, we are interested in computing amplitudes of the from $\mathcal{A}_{\phi\psi}(t) = \langle\phi|U(t)|\psi\rangle$, so that we simply encode forward time evolution in the network, working with tMPS of $N_t = t/\delta t$ sites [2]. We discuss the construction of these objects in Section 3.2.1. In the other case, we are rather interested in computing the expectation value of a given operator, $\langle O(t)\rangle_\psi = \langle\psi|U^\dagger(t)OU(t)|\psi\rangle$, and we typically encode both forward and backwards evolution in our tMPO representation. The most efficient way of doing this is usually to work in a folded representation, that is, building tensors made of the product of one tensor from the forward time evolution operator, together with the corresponding one coming from the backwards evolution. We describe this in more detail in Section 3.2.2.

## 3.1   Models and building blocks

Given a model Hamiltonian $H$, the building blocks of the corresponding temporal MPOs are of course the same as the tensors in the MPO representation of the time evolution operator $U(t) = \exp(-iHt)$. For a Hamiltonian with local interaction, the idea is typically to perform a Trotter expansion introducing small time-steps $\delta t$, breaking up $U(t) = \prod U(\delta t)$ and expressing each $U(\delta t)$ as the product of local operators such as two-body gates, giving rise to the familiar brick-wall structure. One can then further decompose these gates and construct local tensors $W$, as can be seen in Figure 4, ending up with MPOs for $U(\delta t)$ built from rows of local tensors . There are of course several prescriptions for building these tensors, see eg. [12, 14, 15], some of which are implemented in `ITransverse.jl`.

The factorization of $U(t)$ provided by the Trotter expansion allows to build a discrete temporal lattice, where each site is associated with a given instant in time $t = n_t\delta t$. After the clockwise rotation described above, we can view the virtual (bond) links of the MPO as physical *temporal* sites, with a Hilbert space of a local dimension given by the bond dimension of $U(\delta t)$. At the same time, the spatial physical dimension of the initial chain translates into a virtual dimension for the tMPOs we are building.

The user is free to specify the tMPO tensors for whichever additional model they are interested in[3] and use the contraction algorithms described in Section 5. For convenience, in

---

[2]For simplicity, we can think of contracting already the initial and final states in the first and last tMPO tensors, which usually makes life easier if we start/finish with product states.

[3]On a technical note, for the purpose of transverse contraction and truncation on reduced transition matrices, where the hermiticity of the operators involved is not guaranteed, it is often helpful to work at least with *symmetric* objects, so that whenever possible we always prefer to use symmetric left-right tensors (see discussions in Section 5.1 and Section 6.1).

`ITransverse.jl` we also provide helper functions to initialize the tMPO tensors given the required parameters for a few commonly used models. Under the hood, they call the appropriate `build_expH()` function for building the tensors of the $U(\delta t)$ for the model we are interested in. In particular, we provide explicit representations of $U(\delta t)$ for

- The Ising model (with transverse and parallel fields), with Hamiltonian

$$H_{Ising}(g) = -\sum_i \left[ J\sigma_x^i \sigma_x^{i+1} + g\sigma_z^i + h\sigma_x^i \right],\tag{3}$$

with $\sigma_x$ and $\sigma_z$ Pauli matrices.

- The three-state Potts model, defined as

$$H_{Potts}(g) = -\sum_i \left[ J\left( \sigma_i \sigma_{i+1}^\dagger + \sigma_i^\dagger \sigma_{i+1} \right) + g(\tau_i + \tau_i^\dagger) \right],\tag{4}$$

with the matrices $\sigma = \sum_{s=0,1,2} \omega^s |s\rangle \langle s|$, $\omega = e^{i2\pi/3}$ and $\tau = \sum_{s=0,1,2} |s\rangle \langle s+1|$, where the addition is modulo 3.

- The XXZ model, which we can write in terms of the spin operators $S_i$ (so that our implementation works both for spin 1 and 1/2)

$$H_{XXZ} = -J\sum_{i=1}^{L} \left( S_i^x S_{i+1}^x + S_i^y S_{i+1}^y + \Delta S_i^z S_{i+1}^z \right).\tag{5}$$

A particularly useful form for the MPO tensors of $U(\delta t)$ of the Ising model, which is symmetric both in the physical and the virtual legs, is given by the prescription in [12] and implemented in the function

```
build_expH_ising_murg(sites::Vector{<:Index}, mp::IsingParams, dt::Number)
```

where the struct `IsingParams` contains the coupling constants for the model. Its bulk tensors associated with the two-body terms for $J = 1$ are given by

$$W_{Ising}^{bulk} = \begin{bmatrix} \cos(\delta t)\mathbb{1} & \sqrt{-i\sin(\delta t)\cos(\delta t)}\sigma_x \\ \sqrt{-i\sin(\delta t)\cos(\delta t)}\sigma_x & -\sin(\delta t)\mathbb{1} \end{bmatrix}\tag{6}$$

While in principle we could use the same prescription for the Potts model, the resulting tensors do not have left-right symmetry. To get it, we can instead use a symmetric SVD decomposition [43, 44] of the two-body gates (these symmetric decompositions are particularly useful, so we recall them explicitly in Appendix A.1), ensuring that we end up with symmetric tensors. This implementation of the Potts model is implemented in the function

```
build_expH_potts_symm_svd(sites, mp::PottsParams, dt::Number)
```

In addition to these models, for more generic Hamiltonians, we also provide a generic MPO form for $U(t)$ up to second-order Trotter, following the recipe in [15], which can be built starting from the form of the local MPO tensors of a given Hamiltonian[4]. In general this form is not symmetric, but can be still used in our code.

---

[4]Thanks to Jan T. Schneider for sharing the implementation.

## 3.2   tMPO construction and rotation

In `ITransverse.jl` we directly provide helper functions to initialize the tMPO tensors given the required parameters, for the models discussed above. Since we usually rely on Trotterized versions of exponentials of Hamiltonians, we require as input a time-step `dt`, a function `expH_func()` to build the individual tensors (see previous section) and a struct containing the model parameters for the Hamiltonian. When providing these quantities to our constructors, we build the local tensors $W$ for both the real time evolution operator $U(\delta t)$ as well as its imaginary time counterpart, $U(-i\delta t)$. We can specify if we want to incorporate a few steps of imaginary time in our evolution via the `nbeta` parameter. Finally, since often in our calculations we consider translation-invariant initial states that do not change throughout our simulations, we provide the possibility to specify one of these repeating tensors for the initial state, which in our conventions ends up at the bottom end of the tMPO (or, after the 90 degrees rotation described above, at the left edge - hence the "bottom-left / bl " name). All together, the parameters for the tMPO construction are contained in the

```
struct tMPOParams
    dt::Number
    expH_func::Function
    mp::ModelParams
    nbeta::Int
    bl::ITensor  # bottom -> left(rotated)
```

### 3.2.1   Temporal MPO for forward time evolution

Having specified the input parameters, we can use the function `fw_tMPO` to build the tMPO associated with the "forward" time evolution $U(t)$, with the boundaries contracted with the initial and final (spatial) states. For a system which is translationally invariant in space, this is the only ingredient necessary to build the contraction of the full network.

Since we may want to deal with finite systems (or at least use the boundaries of a finite system as initial guess for the power method described later), under the hood we start by constructing the MPO for $U(\delta t)$ for a finite system, and storing the three tensors which will be used as building blocks for all our network, namely the left and right edges of $U(\delta t)$ and the bulk tensor, we shall refer to them as $W_l$, $W_r$ and $W_c$ respectively, see Figure 4. For convenience, we also compute the tensors of $U(-i\delta t)$ associated with imaginary time evolution: these can be used to regularize by cooling down initial states before performing the evolution by replacing the first few $W$ of the tMPO with the corresponding $W_{im}$. These tensors are stored in

```
  struct FwtMPOBlocks
    Wl::ITensor
    Wc::ITensor
    Wr::ITensor
    Wl_im::ITensor
    Wc_im::ITensor
    Wr_im::ITensor
    tp::tMPOParams
    rot_inds::Dict
```

and the `rot_inds` dictionary provides the mapping between the legs of the unrotated and the rotated $W$ tensors, associating the virtual links of the spatial MPO to the physical sites of the rotated tMPO and vice versa.

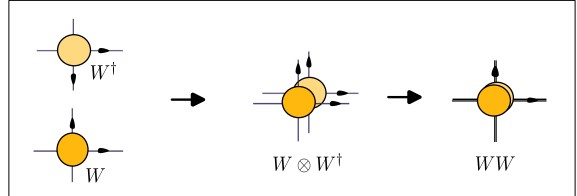

Figure 5: Construction of the folded tMPO tensors, made of pairs $W$ and $W^\dagger$. To build the $WW$ we take their outer product and merge the legs in pairs, resulting in a single tensor which has squared dimensions compared to the original one. For convenience, we draw here some arrows in order to show the ordering of the legs in the product.

### 3.2.2 Folded MPO for expectation values of local operators

As we already mentioned, when dealing with expectation values of local operators it is often convenient to work with folded tMPOs, whose building blocks $WW \equiv W \otimes W^\dagger$ are simply given by the outer product of a $W$ tensor from the forward time evolution contour together with the corresponding one from the backwards evolution path (see Figure 5).

These tensors are constructed and stored in the

```
struct FoldtMPOBlocks
    WWl::ITensor
    WWc::ITensor
    WWr::ITensor
    WWl_im::ITensor
    WWc_im::ITensor
    WWr_im::ITensor
    rho0::ITensor
    tp::tMPOParams
    rot_inds::Dict
```

which has the same structure as `FwtMPOBlocks`, though here we also allow the user to store the (folded) tensor `rho0` associated with the initial state, which can be seen as the local tensor part of a density matrix.

These blocks are built and put together in a folded tMPO by the function

```
folded_tMPO(b::FoldtMPOBlocks, ts::Vector{<:Index}; fold_op, outputlevel)
```

where `fold_op`, which defaults to an identity, specifies the operator we want to put at the folding point of the tMPO between forward and backwards time evolution, while `outputlevel` determines whether additional debugging informations should be printed.

## 4 Truncations and low-level algorithms

Before moving to the high-level algorithms performing transverse contraction, we start by introducing the truncation schemes which are employed underneath.

As customary, most of the transverse contraction algorithms rely on a basic operation, namely the application of tMPOs to left and right tMPS. In turn, a truncation of the bond dimension of the resulting tMPS is required in order to keep the computational cost under control, and we implement in our library several prescriptions tailored for this kind of problem. After briefly reviewing our conventions in Section 4.1 and canonical forms in Section 4.2, we

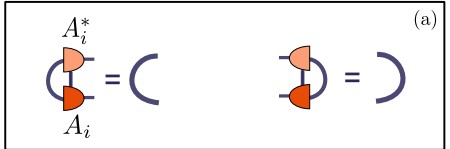 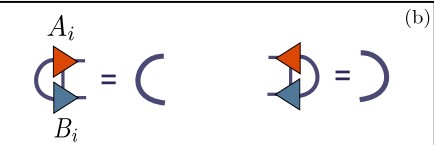

Figure 6: (a): MPS tensors $A_i$ in canonical form reduce to identities when contracted with their conjugate (in the appropriate direction, depending on the form of the isometry, so we can talk about left and right canonical tensors). (b): We can think of a "generalized" canonical form involving two distinct MPS by imposing that the contraction of their tensors $A_i, B_i$ associated with the same physical site reduces to an identity in a similar way. Here and in the following we use the convention that shapes of the same form and in light/darker shades of the same color denote a tensor and its complex conjugate, respectively.

discuss the truncation based on reduced density matrices in Section 4.3 and transition matrices in Section 4.4.

## 4.1 Conventions, overlaps and expectation values

The full contraction of the 2D tensor network associated with the time evolution provides the dynamical quantities we are interested in. If we think of contracting the left half of the system into a single "left tMPS", and the right half into a "right tMPS", the full contraction is simply given by their product.

When thinking about states, we are used to computing overlaps of two "kets" representing vectors $|\psi\rangle$, $|\phi\rangle$ via the scalar product $\langle\phi|\psi\rangle$, which involves the complex conjugation (denoted here with a $*$) going from the ket to the bra. For example, if we take a qubit state $|\phi\rangle = (a, b)$, we will have $\langle\phi| = (a^*, b^*)^T$. In our transverse contraction setup, the quantity we are interested instead is simply given by the product of the two halves of the system, no conjugation needs to be made. We take this into account by considering the left half of the system directly as a "bra" $\langle L|$, whereas the right half is referred to as $|R\rangle$[5]. Their overlap $\langle L|R\rangle$ does not imply thus any conjugation. For this, we provide in ITransverse.jl the function

```
overlap_noconj(L::MPS,R::MPS)
```

which, given $\langle L|$ and $|R\rangle$, computes $\langle L|R\rangle$ without any conjugation[6].

Another scenario which can be of interest is the contraction of the left and right halves of the system into tMPS with the exception of a central tMPO column $E_O$. As an example, it could be that we were able to compute $\langle L|$ and $|R\rangle$ and wanted to use them to compute the expectation value of different local operators on the central site of the system, which could be included in different tMPOs (hence the label "O", as in operator). The full contraction of the network can be written then as $\langle L|E_O|R\rangle$, where again no conjugation is made. We provide the helper function `expval_LR(L,Eo,R)` to compute this overlap in an efficient way.

## 4.2 Canonical forms

Canonical forms are well known to most tensor network practitioners as one of the most powerful instruments allowing to perform local truncations which are globally optimal, as well as to greatly simplify calculations and improve the numerical stability of algorithms (see eg. [4]

---

[5]If the network was symmetric left-right, we would then say that $\langle L| = \langle R^*|$.

[6]This is of course equivalent to `inner(dag(L),R)` using ITensors' `inner()` function, but we do it in a more efficient way which naturally avoids the double conjugation.

$$EIG^{1/2} \left( \text{} \right) = SV \left( \text{} \right)$$

Figure 7: The commonly employed truncation based on reduced density matrices: if we are in the appropriate mixed canonical form, we can access the spectrum of the RDM at a given cut by simply performing a singular value decomposition of the tensor in the orthogonality center, which is equivalent to the eigenvalue decomposition of a similar matrix to the RDM. This in turn allows to perform the optimal truncation (in norm) of the full state by performing a series of local operations, since all the relevant information is contained in the tensor in the orthogonality center. In the case shown here, the orthogonality center is in the central (square) tensor, and the whole RDM spectrum can be obtained by its SVD.

for a review). Given a MPS, we can bring it to a mixed canonical form on a given site by ensuring that all its tensors to the left of a chosen orthogonality center site are left isometries, while those to the right are right isometries. This can be seen as the condition that each MPS tensor contracted with its conjugate in the appropriate direction reduces to a simple identity (see Figure 6). In this form, we can think of the relevant information on the full state as being contained in the orthogonality center tensor, while all the others provide a unitary change of basis from the physical Hilbert space of the full chain into the virtual legs of the MPS. If the orthogonality center corresponds to the first site, the MPS is said to be in right canonical form, whereas it is left-canonical if the center is on the last site.

For later convenience, we can define here a less conventional condition on our tensors, this time involving *two* MPS sharing the same physical sites, which in our case will end up being the tMPS $\langle L|$ and $|R\rangle$. We impose namely that the product of a tensor of the first MPS with the corresponding one of the second, taken again in the appropriate left-right direction, reduces to an identity [23]. In fact, we can even consider the symmetric case $|R\rangle = \langle L^*|$, in which case this translates into a condition on a single MPS. With an abuse of naming, we call tMPS satisfying this condition as being in a *generalized canonical form*[7]

## 4.3 RDM truncation

The simplest algorithm that can be employed for compressing $\langle L|$ and $|R\rangle$ is familiar to each TN practitioner, and involves truncating over the largest eigenvectors of the reduced density matrices built from each tMPS, taken separately. After putting the $\langle L|$ and $|R\rangle$ into the appropriate mixed canonical forms, this is equivalent to performing singular value decompositions (SVDs) on the tMPS tensors, see Figure 7. This truncation, as we all know, provides the best approximation in 2-norm for the $\langle L|$ and $|R\rangle$ vectors individually. Of course, this is a standard truncation used in most MPS libraries, and we can simply call ITensor's `apply()` function to perform it.

One may argue however that this type of truncation is not the best for the problem we are considering: for non-hermitian objects such as the spatial transfer matrices considered here, the result of the optimization turns out to be gauge-dependent. To see it, we can consider the 2D TN made by a series of tMPO columns $E$ sandwiched between the left and right vectors, and apply a transformation (which can be chosen for simplicity to be local) via the tensors $X$, which leaves the network invariant, redefining $E$ as well as $\langle L|$ and $|R\rangle$, see Figure 8. While $E$ is sent into a similar matrix, if we try to build its left and right dominant vectors using their RDM $\rho$

---

[7]Note that this form will be ill-defined if the two vectors are orthogonal.

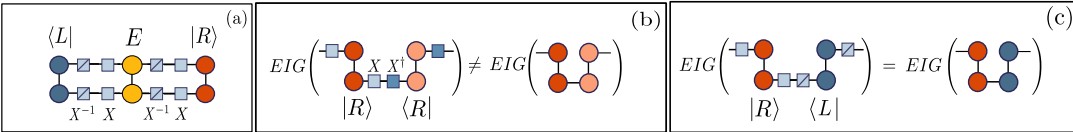

Figure 8: Effects of gauge transformations: we depict in (a) in a schematic way the 2D network we want to contract, represented as the contraction of a left vector with a right one (optionally one can include transfer matrices $E$ in the middle and consider that the network contraction will be given by the overlap of the dominant eigenvectors of $E$). For simplicity and without loss of generality we consider here only two time-steps. We can always perform here a gauge transformation by inserting an identity in the form $XX^{-1}$, with $X$ any invertible matrix, redefining the left and right temporal MPS. (b) While in principle this has no effect on the network contraction, the spectra of the RDM for $\langle L|$ and $|R\rangle$ will depend on this gauge choice. (c) We can see instead that the spectrum of the reduced transition matrix built from $\langle L|$ and $|R\rangle$ is gauge-independent.

$$SV\left(\begin{array}{c}\text{(figure)}\end{array}\right) = EIG^{1/2}\left(\begin{array}{c}\text{(figure)}\end{array}\right) = EIG^{1/2}\left(\begin{array}{c}\text{(figure)}\end{array}\right) = SV\left(\begin{array}{c}\text{(figure)}\end{array}\right)$$

Figure 9: We can compute efficiently the singular values of the RTM if we are in the appropriate canonical form (we refer to Figure 6 for a recap of our graphical notations). As an example, here we start in left canonical form, and show explicitly how - thanks to the properties of the canonical tensors - we can compute the SVD by simply contracting the last tensors of the left and right tMPS.

to perform our truncation, we see that the application of $X$ has a strong effect on their spectra. Crucially, $X$ will in general not be unitary, and in fact can even be seen as some imaginary time evolution which can reduce the entanglement of the boundary states, possibly even turning them into product states. This reduction of rank of the RDM does not translate however into a more efficient representation of the TN: the two states may become increasingly orthogonal as the relevant contributions to their overlap get shifted the tails of singular values which get truncated in the required compression procedure, leading to an inevitable (and uncontrolled) loss of precision in the final result (see [45] for a more detailed discussion of this issue).

## 4.4 RTM truncation

In order to circumvent the limitations of the RDM truncation described above, it can be useful to take a step back and recall that ultimately we are interested in the full contraction of the 2D TN associated with the dynamical quantity we are looking at. This change of perspective suggests that another prescription for truncating the left and right tMPS is possible: one can namely try to find the best approximation for the overlap $\langle L|R\rangle$, rather than focusing on the two vectors individually. A first proposal in this direction was given by [33], where the authors proposed an algorithm[8] for a translation-invariant symmetric system based on truncating over the largest singular values of the matrix $|R\rangle\langle R^*|$ (or equivalently $|L^*\rangle\langle L|$).

---

[8]We sometimes refer to it as the "Hastings" truncation, after the first authors' name.

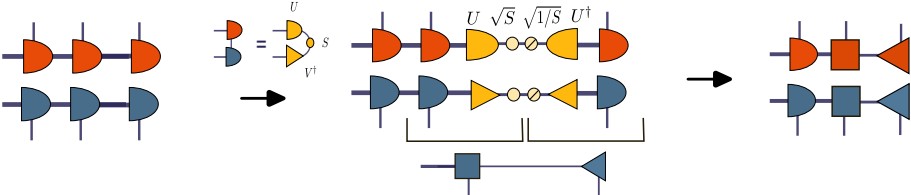

Figure 10: Example of truncation algorithm based on reduced transition matrices, starting from the last sites of the two input tMPS (we refer to Figure 6 for a recap of our graphical notations). We first bring the two tMPS into left canonical form, then perform a right sweep, building environments which at each cut will have the same singular values as the corresponding RTM. At each step of the sweep, we truncate by inserting the appropriate isometries and redefining the tensors of the two tMPS, keeping only the largest singular values. As we move along the temporal chains, we diagonalize the previous environments (possibly even reducing them to identities by appropriately multiplying and dividing by the square roots of the singular values, so effectively bringing the two tMPS in "generalized" canonical form. Here we show explicitly the first step of this truncation with the tensor operations involved, the procedure is then repeated for all sites.

Inspired by that recipe and realizing its connection with the concept of transition matrices, in [23] we developed a truncation algorithm based on reduced transition matrices built from the left and right vectors encoding the full network (including the operators, if we are interested in computing expectation values):

$$\mathcal{T}_t = \mathrm{Tr}_t \frac{|R\rangle\langle L|}{\langle L|R\rangle}\,, \tag{7}$$

which are non-hermitian matrices with unit trace[9]. The final contraction of the full TN will be given by the overlap of the $\langle L|$ and $|R\rangle$ obtained by contracting over the left and right halves of the network.

While ideally we would like to truncate over the eigenvalues of these RTMs, in analogy to what one does with reduced density matrices, non-hermitian matrices will not have a series of positive definite (or even real) eigenvalues, so that a truncation prescription here is not obvious. Instead, in order to provide the best low-rank approximation for our objects we can choose to to truncate on the largest singular values of these RTM, a prescription which is always robust even for non-hermitian objects.

The algorithm, which is sketched in Figure 10, goes as follows [23]: we start by bringing both $\langle L|$ and $|R\rangle$ individually into canonical form in the usual way (we provide the option to do it from left to right and vice versa, which can lead to inequivalent results if the network is not top-down symmetric), let us assume in the following that we bring it into left canonical from. We then begin our truncation: we start constructing the right environment $\mathcal{E}_N$ by contracting together the last tensors of $\langle L|$ and $|R\rangle$, call them $L^{[N]}$ and $R^{[N]}$, where $N$ is the number of temporal sites of the two tMPS. We then proceed to compute the SVD of this environment, which –thanks to the canonical form– corresponds to the SVD of the reduced transition matrix, as can be seen in Figure 9 (where we employed the usual trick that similar matrices have the same eigenvalues, $\mathrm{eig}(UAU^{-1}) = \mathrm{eig}(A)$). We now use the resulting isometries to diagonalize the environment, transforming the tensors $L^{[N]}$ and $R^{[N]}$. In fact, we can even transform

---

[9]In this framework, the algorithm proposed in [33] can be seen as a truncation based on RTM in the symmetric case $\langle L| = |R\rangle$, without the insertion of an operator.

$\mathcal{E}_N$ into an identity by multiplying $L^{[N]}$ and $R^{[N]}$ each by the square root of the inverse of the singular values: in doing so, we bring the left and right tMPS in a form such that the contractions of the $L^{[i]}$ and $R^{[i]}$ to the right of our working site result in identities, see Figure 6 (b). We then proceed along the two tMPS, building environments and diagonalizing them for each temporal site and truncating over their largest singular values.

We can ask ourselves at this point why this particular prescription arises only in the case of the transverse contraction, or, equivalently, to what it would correspond in the case of the usual time evolution à la Schrödinger in the temporal direction. Upon inspecting the network structure, we can see that in that case the "bottom" and "top" vectors which would replace $|R\rangle$ and $\langle L|$ in the construction of the transition matrices are precisely what will end up becoming $|\psi(t)\rangle$ and its conjugate $\langle\psi(t)|$. So in that case the transition matrix would have exactly the same shape as the density matrix for $|\psi(t)\rangle$, so that the two prescriptions (RTM and RDM) coincide.

Both types of truncation are typically controlled by the typical parameters one encounters in a TN calculation: the cutoff below which we discard singular values, and the maximum bond dimension we can afford. For convenience, we pass them around using the

```
struct TruncParams
    cutoff::Float64
    maxbondim::Int64
    direction::String  # "left" or "right"
```

where for the RTM method we also allow the user to specify the `direction` of the truncation sweep, which can lead to inequivalent result for asymmetric networks such as those considered here (see the discussion in Section 5.3).

## 5  High-level algorithms for network contraction

In `ITransverse.jl` we provide two high-level algorithms for performing the transverse contraction of the 2D network associated with the dynamics of our system. The first is a generic power method with determines the left and right dominant eigenvectors of the transfer matrix by repeatedly applying it to starting tMPS. As such, this method works best for a homogeneous system, and it can be applied both for the construction of time-dependent amplitudes as well as the calculation of expectation values. We describe it in Section 5.1. The second method is the light cone algorithm proposed in [46], which is best suited for the calculation of expectation values of local operators in the thermodynamic limit, as we show in Section 5.2. After discussing some caveats and good practices in Section 5.3, we showcase some numerical results obtained using these methods in Section 5.4.

### 5.1  Power method

The power method is a standard algorithm in linear algebra which allows to compute in most cases the dominant eigenvectors of a given matrix, namely those with the largest absolute value (assuming that a gap is present, otherwise the method will be bound to the subspace of degenerate dominant eigenvectors). The idea is simply to start from a trial vector and repeatedly apply the matrix we are interested in diagonalizing to it, the result will increasingly converge to the dominant eigenvector we are interested in. For non-normal matrices, the procedure will need to be done separately for the left and right vectors.

Considering now MPS and MPO as vectors and matrices which we can efficiently multiply, this method provides a way to effectively determine left and right eigenvectors if we are interested in determining the dynamics of a translation-invariant system. This boundary MPS

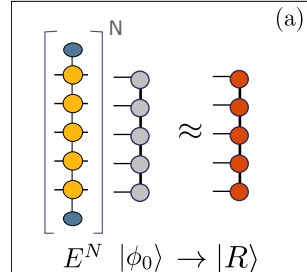 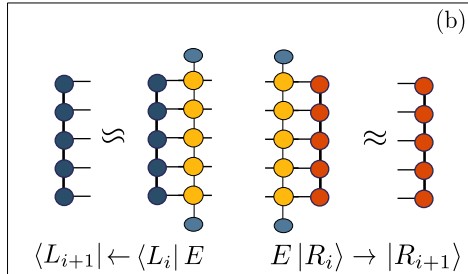

Figure 11: Power method algorithm: for a translation-invariant infinite network, the full network contraction will be given by the overlap of the left and right dominant vectors of one column of the network, which acts as a transfer matrix $E$. (a): By repeatedly applying this tMPO to an initial tMPS $|\phi_0\rangle$, we end up projecting over its dominant vectors, with a convergence rate dictated by its gap. (b): At each step of the power method we apply the tMPO $E$ to the left and right tMPS, and re-compress them, repeating until convergence is reached (see main text for details).

method is commonly employed in the contraction of 2D networks, and we present here our implementation tailored for the contraction associated with the dynamics. In practice, for most cases, as long as we evolve for a finite number of time-steps the transfer matrices we consider have a gap [24], guaranteeing a relatively quick convergence of the method.

Let us now briefly describe the actual implementations provided in `ITransverse.jl`. The power method parameters are controlled via the

```
struct PMParams
        truncp::TruncParams
        itermax::Int64
        eps_converged::Float64
        increase_chi::Bool
        opt_method::String
```

The power method runs for a maximum of `itermax` iterations, or stops earlier if convergence has been reached. The latter is estimated as the variation of the entropy $\Delta S$, which is computed via RDM or RTM, depending on the optimization method used (as determined by `opt_method`, see below and Section 4) between the tMPS computed in subsequent iterations of the power method: if $\Delta S <$ `eps_converged`, we say that the power method has converged[10]. In some cases, the bond dimension reached in the intermediate steps of the power method is significantly larger than the one required for the final converged tMPS. In fact, if we have a good guess on the bond dimension required for the final result, it is sometimes possible to specify a `maxbondim` close to that for the algorithm, which in this case will typically need more iterations to converge, but each one at a lower computational cost. In any case, we can try to slow down the growth of this bond dimension in the hope that convergence is reached before larger dimensions are hit: this is governed by the `increase_chi` parameter, which -if set to true- limits the maximum bond dimension in the initial iterations of the power method, while letting it gradually increase towards the end.

We provide different functions to perform the power method, depending on the problem at hand. The first option is provided by the function

```
powermethod_both(in_mps::MPS, in_mpo_L::MPO, in_mpo_R::MPO,
                 pm_params::PMParams; flip_R::Bool=false)
```

[10]Other options, such as considering the fidelity between the tMPS in a step and that of the following one are possible, of course.

which, starting from an initial guess `init_mps` on both sides, at each step applies a column `in_mpo_L` to the left and a `in_mpo_R` to the right, then performs the truncation either individually on the new $\langle L|$ and $|R\rangle$ using RDM, or optimizing the RTM $|R\rangle\langle L|$, as described above. So both left and right vectors are optimized, hence the name.

Whenever possible, working with systems that have a left-right symmetry with respect to the center allows to greatly simplify the contraction procedure, since we only need to update one of the two tMPS and obtain the other by simple transposition. We implement the symmetric power method for this case in the function

```
powermethod_sym(in_mps::MPS, in_mpo::MPO, pm_params::PMParams)
```

which takes as input a single MPS `in_mps`, applies the `in_mpo` to it and truncates the resulting MPS, repeating until convergence is reached, as before. We can specify in `pm_params` the `opt_method = RDM` if we want to use the usual truncation based on the reduced density matrix, or `opt_method = RTM`, in which case, since the problem is symmetric, we compute the SVD of the symmetric environments in a Autonne-Takagi form (see Appendix A.1). In fact, in this symmetric case one can also compute efficiently the (complex!) eigenvalues of the RTM (we discuss this in more detail in Section 6.1), so we also provide an experimental truncation which discards the eigenvalues with the smallest modulus, which can be selected via `opt_method = RTM_EIG`.

Suppose now that we are interested in the contraction of a network associated with the expectation value of a local operator in an infinite system. In the transverse picture, we can see the final result as the contraction $\langle L_{-N}|E_1^N E_O E_1^N|R_N\rangle$, for $N\to\infty$ so that the initial guesses for the boundaries $\langle L_{-N}|$ and $|R_N\rangle$ are not important. Here $E_1$ is a tMPO column without any operator, while $E_O$ denotes the column(s) containing additions of local operators between the forwards and backwards contour. For this kind of setup, we always work in the folded picture, which provides the most efficient representation of our problem.

Here one might be tempted to optimize left and right vectors without the central column $E_O$, and include it only at the end to evaluate the expectation value. This however will not work: without an operator, the contraction of the folded tensors reduces to a series identities, in fact the whole network would just reduce to the trivial overlap $\langle\psi_0|U^\dagger(t)U(t)|\psi_0\rangle = \langle\psi_0|\psi_0\rangle$. The RTM-based truncation would then return trivial product states for $\langle L|$ and $|R\rangle$, losing all the information on the dynamics.

Instead, one can use the prescription implemented in the function

```
function powermethod_op(in_mps::MPS, in_mpo_1::MPO, in_mpo_O::MPO,
            pm_params::PMParams)
```

which goes as follows: for each step, starting from the current $\langle L|$ and $|R\rangle$ we build an updated left tMPS $\langle L|_1$ by applying `in_mpo_1` to it, and a right one $|R_O\rangle$ by applying `in_mpo_O` to the right. The resulting transition matrix $\mathcal{T}\sim|R_O\rangle\langle L_1|$ is then used to optimize $\langle L_1|$, which is taken as the new $\langle L|$, while $|R_O\rangle$ is discarded. The same procedure is then repeated the other way around: we build $\langle L_O|$ and $|R_1\rangle$, optimize their RTM and obtain the updated $|R_1\rangle$, which will be the new $|R\rangle$. We then repeat until convergence is reached. This method, which works for a generic case without any left-right symmetry, is employed if we pass the `pm_params.opt_method=RTM_LR` parameter.

If the tMPOs are symmetric left-right, we should be able to perform the update only on one of the two vectors, say $\langle L|$, and obtain immediately $|R\rangle$ by transposition[11]. This prescription is used if we pass the `pm_params.opt_method=RTM_R` parameter.

Finally, for convenience we also allow here to perform the power method by using the common truncation based on the RDM of $|R\rangle$ at each iteration. In this case, which is used

---

[11]Note that this is *not* the same as performing a symmetric update of the form $\langle R|R\rangle$, here we still update the overlap $\langle LO|1R\rangle$

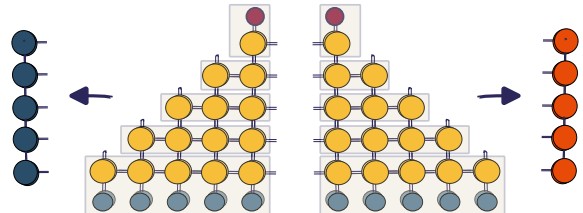

Figure 12: After transverse contraction, the left and right halves of the light cone TN can be represented as tMPS, which can be built iteratively for each timestep, as discussed in the main text.

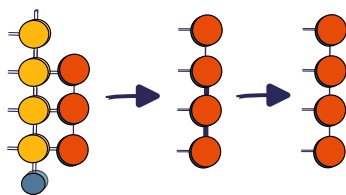

Figure 13: Updating the right tMPS using the light cone algorithm. At each step, we apply to the current tMPS a tMPO which has one more temporal site on top (built from the tensor $W_r$), and obtain a new tMPS with one additional site and larger bond dimension. We then proceed to truncate the tMPS. In this way, the tMPS for $N_t$ time-steps can be obtained by the one for $N_t - 1$ steps with just the application of a single tMPO.

when `pm_params.opt_method=RDM`, the `in_mpo_0` input is unused, and the power method is performed using `in_mpo_1` only.

Since the tMPOs are usually not unitary operators, at each step of the power method we normalize back the tMPS, in order to avoid losing precision over many iterations. The most consistent way to do it is to enforce that the overlap $\langle L|R \rangle = 1$, but in practice normalizing individually $\langle L|L \rangle = \langle R|R \rangle = 1$ works as well. Another possibility is to normalize the overlap $\langle LO|1R \rangle$ before truncating for $|Rnew\rangle$.

## 5.2   Light cone

Initially proposed in [46] (see also [47]), the light cone method provides an efficient way of building the left and right tMPS associated with the expectation value of a local operator. The idea behind this iterative method is precisely to exploit the causal (or "light") cone associated with such operator to work with a significantly reduced number of tensor [48], see Figure 12. One can start with an initial set of left and right tMPS $\langle L_1|$ and $|R_1\rangle$ for time $t_1$, which can be given by a single tensor if we start from $t_1 = \delta t$, and from there build the tMPS associated with the time evolution for all subsequent times. In practice, in order to build the tMPS $\langle L_{i+1}|$ and $|R_{i+1}\rangle$, one applies to them a tMPO with an additional time site, which effectively extends them (Figure 13):

$$|R_i\rangle \ \rightarrow \ |R_{i+1}\rangle = E_{i+1}|R_i\rangle \,, \quad \langle L_i| \ \rightarrow \ \langle L_{i+1}| = \langle L_i|E_{i+1} \,, \qquad (8)$$

then a truncation is performed, in a similar way to the power method. The advantage of the light cone is that, given the tMPS describing the folded left and right vectors at a time $t_1$, we are able to build from them the tMPS at a later time $t_2 > t_1$ constructively from them by applying layers of tMPO, without needing to re-contract the full network until convergence.

Given the causal structure dictated by the local operator, the expectation value obtained will match the value in the thermodynamic limit. In its simplest version, for each additional timestep the algorithm extends the system by one spatial site, allowing for a description of physical velocities up to $v_{max} = 1/\delta t$. In fact, for most cases this prescription is excessive, as physical Lieb-Robinson velocities usually fall below this value. One can then work with narrower light cones, extending the cone in space only every few timesteps [46].

In `ITransverse.jl` we have implemented the function

```
init_cone(tp::tMPOParams, n::Int)
```

which initializes the tMPS for `n` timesteps and a given set of `tMPOParams`, without performing any truncation. The function

```
run_cone(psi::MPS, b::FoldtMPOBlocks, cp::ConeParams, nT_final::Int)
```

then evolves the input tMPS `psi` until `nT_final` timesteps, with the time evolution MPO built from the folded blocks contained in the struct `b`.

The parameters of the algorithms are contained in the

```
struct ConeParams
    truncp::TruncParams
    opt_method::String
    optimize_op::Vector
    which_evs::Vector{String}
    which_ents::Vector{String}
    checkpoint::Int
    vwidth::Int
```

where `optimize_op` allows to define the operator we want to optimize for, using the same strategy as in the function `powermethod_op()` described above, ie. truncating using RTMs containing the operator (in fact, the parameter `opt_method` is the same as for the power method). Since the algorithm evolves in time step by step, we can compute the expectation value of local operators and (generalized) temporal entropies at each iteration to follow their time evolution. We allow the user to specify which quantities to compute via the parameters `which_evs` and `which_ents`. Finally, we allow to account for physical velocities smaller than $1/\delta t$ via the `vwidth` parameter (defaulting to 1), which provides the number of timesteps after which the cone should be extended by one spatial site.

## 5.3 Caveats and general considerations

We can see the algorithms presented here as falling into two groups: the first one is tailored for systems with a simple structure, like the Loschmidt echo for a translation invariant system, where the whole network is basically described by a single column which is repeated indefinitely. In that case, the strategy for performing updates is straightforward: we simply apply one of these columns to the left and the right, and then subsequently truncate. As we argued in the above sections, the best truncation here should be based on transition matrices.

The second group is tailored to problems which, in a way or another, break translation invariance, such as the expectation values of local operators. Take as an example the light cone for an operator , which is built step by step by inserting longer tMPO columns in the center, "pushing" the previous tMPS outwards by one site, that is, we are growing the system from the center outwards. A central observation made in [23] is that, if we begin our truncation sweep from the side of the operator at a given step, we can directly connect with the operator space entanglement of the time-evolved $O$. It turns out however that this truncation is not a good basis for the light cone algorithm described above: the issue is that for each time-step, we are optimizing the tMPS associated with the expectation values of the local operator at

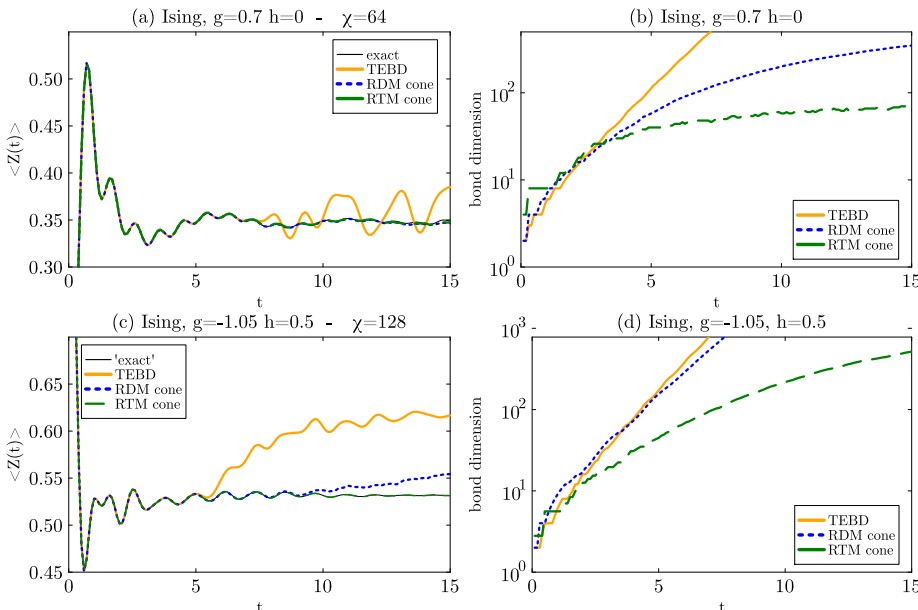

Figure 14: Comparison between traditional (TEBD) and transverse contraction methods (both RDM and RTM-based, see main text) for an infinite Ising chain. (a): Expectation value of $\sigma_z$ at mid-chain for an integrable Ising model, with a maximum bond dimension of 64: we see that TEBD starts deviating from the exact result already around $t \approx 8J$, while transverse contraction methods faithfully reproduce the correct value. (b): Bond dimension required by the algorithms for an integrable Ising model with a fixed cutoff $10^{-8}$. All transverse contraction algorithms exhibit a logarithmic scaling of the bond dimension, hinting at efficient simulability of the dynamics with tMPS, compared to the exponential scaling of TEBD. (c): Expectation value of $\sigma_z$ for non-integrable Ising with transverse and parallel fields, enforcing a maximum bond dimension $\chi = 128$. Again we see that transverse contraction methods, particularly those based on RTM truncation, are able to capture the correct results with a limited $\chi$, whereas TEBD fails already around $t = 5J$. (d): Bond dimension for non-integrable Ising. Here RDM-based transverse truncation also exhibits an exponential growth, while the RTM truncation seems to show a more favorable scaling.

the given time we are considering. There is however no guarantee that the optimized tMPS which will give the correct expectation value at a time $t$ can be used as good starting point for building the tMPS for a system at $t' > t$. In other words, we might be truncating over a non-optimal quantity, discarding information which will be required at later steps. A workaround to this is to truncate starting from the side of the initial state by setting `direction="left"` in `TruncParams`. In the absence of an operator and for a symmetric case, this boils down to the truncation proposed in [33] and employed already in [46].

## 5.4   Some numerical results

While we defer a more complete discussion on the computational complexity of the transverse contraction to the next section, as an appetizer let us showcase here some numerical results of the algorithms for the simple case of the expectation value of a local operator, built using the light cone. We will compare both RDM and RTM truncations with the results obtained with a standard TEBD algorithm, taken as prototype of the traditional time evolution methods which evolve the wave-function in time.

In Figure 14 we show results for both the integrable transverse field Ising model, as well as the non-integrable version with parallel field (Equation (3)). For the latter, we choose a parameter set which should describe a chaotic dynamics, presenting a considerable challenge for the study of time-evolution.

We start by showing the expectation value of the $\sigma_z$ operator at mid-chain, compared to the exact value[12] in the various algorithms for a fixed maximum bond dimension, showing that transverse methods provide a much superior performance.

We also compare the bond dimensions required by the algorithms if we impose a cutoff of $10^{-8}$ on the truncated singular values. The difference is striking in the integrable case (b), where we see that the bond dimension $\chi$ required by the light cone algorithm with RTM truncation exhibits a logarithmic growth, remaining below $\chi = 100$ even beyond $t = 10J$. The RDM truncation also seems to have a logarithmic behavior, albeit with a larger prefactor, in contrast with the exponential growth of regular TEBD (note the log scale). The non-integrable case (d) looks qualitatively different: the bond dimension for transverse methods based on RDM truncation scales exponentially, whereas the RTM-based method also grows significantly but seems sub-exponential.

Many other numerical results obtained with `ITransverse.jl` can be found in the literature [23–27]. We encourage interested users to check the `examples/` folder in the package repository, where we provide several scripts to play around with different models and truncations.

# 6 Temporal entropies and computational complexity of the transverse contraction

After presenting all the tools for transverse contraction, it is now time to discuss when these algorithms can be more useful compared to the tried and true Schrödinger and Heisenberg evolutions. While in general the question is still open, there are already several results which provide some insight in this direction, and we briefly review them in this section.

We need not stress here the relevance of the concept of entanglement entropy and all its implications. From the point of view of a tensor network practitioner, one of the most important relationships is the one between the entropy of a state and bond dimension required for its faithful description: the higher the entanglement, the larger the bond dimension. In turn, this means that we can only represent efficiently states with small entanglement: for one-dimensional chains we usually think about states fulfilling an area law (where the entanglement, and thus the bond dimension, remain bounded), or with at most logarithmic violations to it, corresponding to a polynomial resource cost in terms of memory for the tensors.

In this sense, the entanglement entropy of a state, built from its reduced density matrices, is a clear indicator of the computational complexity of representing faithfully that state using MPS. A natural construction for our case is then to compute the entropy of left and right temporal states, and expect that such a *t*emporal entropy will provide an estimate of the cost of computing the dynamical properties we are interested in - after all, the final result will be given by the overlap $\langle L|R \rangle$.

We have seen already however in Section 4.3 that for our case one has to be careful, and even if the temporal states taken individually can be deceivingly simple, their overlap may be not accurately reproduced. We can argue instead that the proper object reflecting the complexity of representing the dynamics is encoded in the transition matrices defined above, which are built from the two vectors together [23].

---

[12] For the non-integrable case we treat as "exact" the value obtained with TEBD using the largest bond dimension we were able to use, $\chi = 4096$ in our case.



Figure 15: We can compute the generalized Renyi-2 entropy efficiently by simply constructing two copies of the RTM (L1,R1 and L2,R2, respectively) and contracting them appropriately. Here we draw schematically the TN contraction for $\mathrm{Tr}\left(\mathcal{T}_t^2\right)$ for a cut in the middle of the temporal chains.

Let us then introduce more formally the concept of generalized temporal entropies, and show and how can they be calculated using `ITransverse.jl`.

## 6.1 Generalized temporal entropies

The idea of generalized entropies has been proposed in recent high-energy physics literature [36–38, 49] discussing transition matrices between different states. In our case, these will be the left and right temporal states, so that we can talk about generalized temporal entropies [38, 39, 42]. If we now consider, for example, the generalized Von Neumann entropy,

$$S_1^{gen}(t) = -\mathrm{Tr}\,\mathcal{T}_t \log \mathcal{T}_t,$$

we usually think about this expression in terms of the eigenvalues of the RTM $\mathcal{T}_t$. Assuming that this non-hermitian matrix can be diagonalized, these eigenvalues will in principle be complex, leading in turn to complex-values entropies.

From a computational point of view, there is an additional complication here: the RTM are exponentially large matrices in the number of time-steps of the time evolution, so that exact diagonalization is obviously unfeasible for them beyond very short times. For the calculation of reduced density matrices, the MPS machinery allows for an efficient diagonalization thanks to the canonical form, which greatly simplifies the problem: the singular values extracted from an MPS tensor in the orthogonality center are precisely the square roots of the eigenvalues of the RDM at that cut. Unfortunately, we cannot use straightforwardly the same trick for a RTM, built from two different vectors: even after a gauge transformation in a "generalized" canonical form, the tensors on the two sides will be different, so that we cannot rely on a similarity transformation to simplify our problem here.

One exception to this is given once again by the symmetric case, in which $\langle L| = |R\rangle$. There, we can diagonalize the RTM with (complex) orthogonal matrices [50] (see Appendix A.2), so that we are able to build an environment which is similar (ie. has the same eigenvalues) to the RTM we want to diagonalize.

This symmetric diagonalization is implemented in the function

```
diagonalize_rtm_symmetric(psiL::MPS;
        bring_left_gen::Bool=true, normalize_eigs::Bool=true,
        sort_by_largest::Bool=true, cutoff::Float64=1e-12)
```

Even if we cannot diagonalize RTMs directly in the non-symmetric case, we can still define higher-order entropies, which can be computed efficiently using MPS. The first obvious examples are the (generalized) purity and the $\alpha = 2$ Renyi entropy, Figure 15

$$S_2 = -\log Tr\left(\mathcal{T}_t^2\right). \tag{9}$$

The evaluation of these generalized entropies is provided by the functions

```
gen_tsallis2(psi::MPS, phi::MPS)
gen_renyi2(psi::MPS, phi::MPS)
```

which, given two MPS `psi` and `phi`, build the appropriate contraction at each temporal cut and compute efficiently the required traces.

## 6.2 Computational complexity

Now that we introduced the definition of generalized temporal entropies, we can finally ask the question on what is the computational complexity of the transverse contraction algorithms described here.

There are several works in the literature studying the scaling of the "standard" temporal entanglement which comes into play in the RDM truncation, particularly in the context of discrete (Floquet) dynamics, see eg. [22, 46, 51–55].

Let us discuss here instead the scaling of generalized temporal entropies, which - as we argued - should be more closely related to the dynamical problems we are interested in here, such as the evaluation of time-dependent expectation values and amplitudes.

The RTM truncation described in Section 4.4 allows for a first connection to determine the computational cost of computing expectation values of operators. By performing this truncation procedure, one can see that the rank of the RTM involved in the contraction is upper bounded [23] by the rank of the matrix encoding the operator space entanglement entropy [56]. In the worst case scenario, this is still a quantity that could exhibit a linear growth with time [57, 58], ie. a volume law corresponding to an exponential entanglement barrier.

In order to get a better intuition on the computational complexity of the problem and the role of generalized temporal entropies, it is useful to take a step back and focus on a simpler TN, namely the one associated with a time-dependent amplitude such as a Loschmidt echo (Figure 1). In [24], this kind of setup for a quench of an infinite system evolved with a critical Hamiltonian was mapped to a path integral on a strip, allowing to exploit the machinery of conformal field theory (CFT) to predict the behavior of generalized temporal entropies. As long as an analytical continuation from euclidean to real time evolution can be performed, the growth of these entropies in this context dictates the compressibility of the relevant left and right temporal MPS, much in the same way as what happens for the spatial entanglement of critical ground states. Furthermore, CFT predicts a logarithmic growth of generalized entanglement at criticality, hinting at a polynomial cost and thus efficient simulability of amplitudes using tMPS.

Recently, a numerical exploration beyond this analytical regime has shown that the generalized temporal entropies still exhibit a logarithmic growth after crossing dynamical quantum phase transitions - that is, times at which physical quantities exhibit discontinuities [59, 60]. Hinting at some universal properties for large times, this suggests that there is no change in complexity related to these transitions [27], and that transverse contraction methods might provide access to the long-time dynamics of quantum systems in an efficient way.

Finally, we recall that, strictly speaking, for a general non-symmetric case the RTM truncation is based on a singular value decomposition. The rank of the objects involved in that case could then be related to the concept of *SVD entropies*, which have also been recently discussed in the literature [61, 62]. A detailed study of these quantities would also be of extreme interest, and could possibly provide more rigorous bounds for the computational complexity of our methods.

# 7   Conclusion

Transverse contraction methods provide an efficient way to study the dynamics of quantum many-body systems with tensor networks, allowing (at least in some cases) to circumvent the exponential entanglement barrier associated with the conventional Schrödinger and Heisenberg time evolutions. By considering the $D+1$-dimensional tensor network associated with the temporal evolution of a $D$-dimensional system and introducing boundary "temporal" states, the network contraction can be performed in the spatial direction. The complexity of the contraction for these boundary MPS methods is related to the (generalized) temporal entanglement, built from reduced transition matrices. For one-dimensional quantum systems, there is evidence that many temporal states associated with computing time-dependent amplitudes satisfy an area law for this generalized entanglement, or at most a logarithmic growth, hinting at the possibility of an efficient representation using temporal matrix product states. This efficient representation however does not come completely for free: the typical objects entering into transverse contraction algorithms for time evolution are complex and non-hermitian, so that additional care must be taken for the numerical stability and reliability of the algorithms.

We presented here the `ITransverse.jl` julia package, which provides state of the art algorithms based upon the transverse contraction for the dynamics of one-dimensional quantum many-body systems. The package provides both high-level routines to perform the full dynamical calculation of expectation values and amplitudes, including a power method and a light cone algorithm, as well as several functions to efficiently perform truncations based on reduced transition matrices and compute generalized entanglements. All these functionalities make it a powerful toolbox, which allows to study different numerical problems in an efficient way using these novel algorithms.

While the algorithms presented here are meant for the time evolution of one-dimensional systems, many of the ideas reviewed here can be extended to higher-dimensional problems - in fact, some initial works in this direction have already been presented in this direction [63]. We look forward to implementing these methods in `ITransverse.jl` in the near future.

## Acknowledgements

The algorithms included in `ITransverse.jl` have been developed to investigate all sorts of physical questions related with the dynamics of quantum many-body systems. Most of this work has been performed in close collaboration with Luca Tagliacozzo, together with Aleix Bou-Comas, Jan T. Schneider, Sergio Cerezo Roquebrún, Esperanza López, Jacopo de Nardis, Guglielmo Lami and Carlos Ramos Marimón. Special thanks to Luca Tagliacozzo, Aleix Bou-Comas, Jan T. Schneider and Mari Carmen Bañuls for helpful discussions and feedback on the manuscript.

**Funding information**   SC acknowledges his AI4S fellowship within the "Generación D" initiative by Red.es, Ministerio para la Transformación Digital y de la Función Pública, for talent attraction (C005/24-ED CV1), funded by NextGenerationEU through PRTR.

## A   Symmetric Singular value and Eigenvalue decompositions

The transition matrices which appear in transverse contractions are typically complex and non-hermitian operators, requiring extra care in their treatment. Luckily, in some cases we can consider models with a left-right symmetry in the corresponding MPO tensors - in this

case, these transfer matrices are symmetric, ie. $E = E^T$ (where $E^T$ denotes only transposition, without complex conjugation), a property which allows us to perform a few very useful decompositions, as described below.

## A.1 Symmetric SVD

A complex symmetric matrix allows for a special singular value decomposition reflecting this symmetry, often referred to as an Autonne-Takagi decomposition [43,44]. Given a symmetric matrix M, we can start by performing the standard SVD $M = USV^\dagger$, truncating as usual over the smallest singular values if necessary. From this, we construct the (block-diagonal, or diagonal if there are no degenerate singular values) matrices $Z = U^\dagger V^* = V^\dagger U^* = Z^T$, and from these and $U_Z = U\sqrt{Z}$. This allows us to reach the following decomposition using an isometry and its transpose

$$M = U_Z S U_Z^T \,, \tag{A.1}$$

## A.2 Symmetric eigenvalue decomposition

Symmetric matrices also allow for an eigenvalue decomposition using (complex) orthogonal matrices [50], namely

$$M \approx O\Lambda O^T \,, \quad O^T = O^{-1} \,. \tag{A.2}$$

To get this form, we first obtain the right eigenvectors $V_R$ of M, $MV_R = V_R\Lambda$, from which we can build the transformation matrix $O$ by first constructing the block-diagonal matrix $G = (V_R^T V_R)$ and then computing $O = V_R G^{-1/2}$.

   This decomposition is particularly useful as it defines a similarity transformation which allows us to diagonalize the reduced transition matrices in a symmetric case using low-rank representations, as discussed in Section 6.1.

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
