# Peer review of "The ITransverse.jl library for transverse tensor network contractions"

_SciPost Physics Codebases_

## Round 1 · Referee Report · Atsushi Ueda (Referee 1) · 2025-11-13

Strengths
Very high quality paper. It provides a wide range of examples for applications. This package also provides a great step to the author's more recent paper on time-evolution with sampling. Definately one of the most state-of-art package for time-evolution in the community.
Weaknesses
NA
Report
This paper introduces a user-friendly Julia package for simulating time-evolution of one-dimensional quantum systems. Conventionally, time-evolution of quantum states are performed through applying a time-evolution operator on a state, TEBD and TDVP, for instance. These schemes are known to suffer from so-called "entanglement barrier," which refers to the exponential growth of entanglement with time. As the maximum entanglement in the simulation is fundamentally restricted by the bond dimension of MPSs, the entanglement barrier was inevitable limitation to simulate long-time dynamics.
This paper instead represents a time-evolution of the states by evaluating a 2d space-time tensor network. The main advantage of this view is that contracting the tensors in the space direction often can circumvent the entanglement barrier. This allows us to access the quantum dynamics way beyond the traditionally achievable time range. The package provides a several intuitive way for simulating this state-of-art technique, opening up the new avenue for simulating quantum quenches and such.
The paper provides extensive explanation of both theoretical background and hands on benchmarking codes(also lots thereof in the github repo). I believe this paper matches all the criteria for this journal.
This paper instead represents a time-evolution of the states by evaluating a 2d space-time tensor network. The main advantage of this view is that contracting the tensors in the space direction often can circumvent the entanglement barrier. This allows us to access the quantum dynamics way beyond the traditionally achievable time range. The package provides a several intuitive way for simulating this state-of-art technique, opening up the new avenue for simulating quantum quenches and such.
The paper provides extensive explanation of both theoretical background and hands on benchmarking codes(also lots thereof in the github repo). I believe this paper matches all the criteria for this journal.
Requested changes
I am very happy with the current manuscript.
Recommendation
Publish (surpasses expectations and criteria for this Journal; among top 10%)

---

## Editorial Decision

voting_in_preparation